**Subject Category:**
Biology (whole organism)

neuroscience/physiology/developmental biology

antennal lobe, mushroom body, calyx, butterfly, olfaction, plasticity

**Author for correspondence:**
Mikael A. Carlsson
e-mail: mikael.carlsson@zoologi.su.se

# Insect brain plasticity: effects of olfactory input on neuropil size

Maertha Eriksson, Sören Nylin and Mikael A. Carlsson

Department of Zoology, Stockholm University, 106 91 Stockholm, Sweden

 ME, 0000-0002-9820-847X; SN, 0000-0003-4195-8920;
MAC, 0000-0002-9190-6873

Insect brains are known to express a high degree of experience-dependent structural plasticity. One brain structure in particular, the mushroom body (MB), has been attended to in numerous studies as it is implicated in complex cognitive processes such as olfactory learning and memory. It is, however, poorly understood to what extent sensory input *per se* affects the plasticity of the mushroom bodies. By performing unilateral blocking of olfactory input on immobilized butterflies, we were able to measure the effect of passive sensory input on the volumes of antennal lobes (ALs) and MB calyces. We showed that the primary and secondary olfactory neuropils respond in different ways to olfactory input. ALs show absolute experience-dependency and increase in volume only if receiving direct olfactory input from ipsilateral antennae, while MB calyx volumes were unaffected by the treatment and instead show absolute age-dependency in this regard. We therefore propose that cognitive processes related to behavioural expressions are needed in order for the calyx to show experience-dependent volumetric expansions. Our results indicate that such experience-dependent volumetric expansions of calyces observed in other studies may have been caused by cognitive processes rather than by sensory input, bringing some causative clarity to a complex neural phenomenon.

## 1. Introduction

Structural plasticity of the adult brain is well documented and is a prerequisite to cope with an ever-changing environment. As brain tissue is energetically expensive to maintain [1], structural plasticity regulated by environmental conditions is an effective way of keeping the brain as small and inexpensive as possible while still allowing certain neuropils to expand in order to meet induced needs for specific processing power. For example, fluctuations of the relative size of the hippocampus in birds are correlated with food storing [2] and the brains of mice living in an enriched environment differed volumetrically from the

brains of mice that lived under normal laboratory conditions [3]. Likewise, certain brain structures in adult insects have demonstrated remarkable plasticity associated with learning, food search and novel tasks [4–8]. However, the direct causes of structural brain plasticity are still poorly understood.

Most insects depend on olfactory cues for finding resources such as food, mates and host plants, as well as for avoiding potential dangers. Because of this, brain regions dedicated to processing olfactory information often make up a significant portion of the brain. The primary neuropil receiving olfactory input among insects is called the antennal lobe (AL). Here, olfactory information is categorically sorted according to the chemical properties of detected odorants [9–12] in an organizational manner which resembles that of mammals [13]. Being far from a passive relay station for olfactory input, the AL contains an intricate web of local interneurons and extrinsic centrifugal neurons which allow substantial signal processing and modulation within the neuropil [14–20]. Volumetric plasticity in ALs has been observed in a wide range of insects [5,21–25], and a general finding among such studies is that ALs tend to increase in volume as insects grow older and gain new experiences.

AL projection neurons (PNs) primarily forward olfactory information from the AL to two higher neuropils, the mushroom body (MB) and the lateral horn (LH). While the main function of the LH appears to revolve around categorical sorting of olfactory information according to ecological relevance [10,26–30], the MB is involved in more complex cognitive functions [31–33]. Although the bulk of presynaptic input to the MB generally is of olfactory nature, it may also receive input from visual, tactile, auditory and gustatory sensory sources, making it an important centre for context generalization and integration of information from different sensory modalities [34–42]. Furthermore, the MBs are involved in higher cognitive functions, such as associative and spatial learning, memory storage and decision making [32,33,35,41,43,44].

Many studies show that experiences that challenge the processing power of senses and functions associated with the MB often cause volumetric changes to the MB input region (calyx) and/or output region (MB lobes) [5,8,21,22,24,45–48]. Although the experimental design varies between these studies, all have in common that treatments which affected MB plasticity included some measure of behavioural freedom for the animals. As a consequence, it is difficult to conclude to which extent volumetric changes of the MBs were caused by sensory input or generated by complex behaviours, different types of learning, or other cognitive processes.

It was recently demonstrated that host plant search in butterflies dramatically affected the plasticity of their MBs [8]. In that study, it was further shown that increased complexity of the host environment was positively correlated with MB calyx volume. However, as the experimental design allowed free movement and access to a variety of plants, it was not possible to disentangle the variables causing the calyx expansion.

In the present study, we used selective sensory deprivation to investigate the involvement of olfactory input to the MBs as a source of the structural plasticity previously demonstrated in the comma butterfly, *Polygonia c-album* [8]. We promoted olfactory input by providing an odour-rich environment, but denied any form of physical interaction between insect and odour source. This set-up aimed to eliminate most confounding factors derived from insect behaviour, associative learning and spatial navigation, as well as from other types of sensory input such as tactile and gustatory.

# 2. Material and methods

## 2.1. Species and rearing

A stock population of *Polygonia c-album* (Linné, 1758) originating from adults collected in the Norfolk region of eastern UK, provided by the commercial company Worldwide Butterflies (UK), was used in this study.

Eggs from approximately 60 females were pooled, and newly hatched larvae were transferred to plastic cups in groups of five. Larvae were kept at 17°C on a short-day–light cycle (12 : 12 L : D) until reaching third instar, and thereafter kept at 23°C on a long-day–light cycle (22 : 2 L : D) until pupation. All larvae were fed fresh cuts of stinging nettle (*Urtica dioica*) ad libitum. Pupae were sexed by inspection of genital slits on the day after pupation, and kept at 17°C until adult eclosion. Only female butterflies were used in this experiment. Females used in the tracing experiment and the olfactory blockage experiments were different individuals but from the same population.

## 2.2. Tracing experiment

Newly eclosed butterflies were anaesthetized on ice and placed in cut pipette tips to allow the head to protrude at the narrow end. An opening in the cuticle was done at the sites covering either the medulla, the lobula complex, the calyx of the MBs or the ALs. A glass electrode coated with vaseline and neurobiotin (NB) crystals (Vector Laboratories, USA) was transiently injected into the neuropil of interest whereafter moth saline was applied [49] and the animal was left overnight in a humidified chamber at 4°C. The next day, the brain was dissected out and the same protocol for fixation and incubation (anti-synapsin) as in [8] was used, except that Alexa Fluor 488 conjugated streptavidin (1 : 500, Life Technologies, The Netherlands) was added to the Alexa 546-tagged secondary antibody.

## 2.3. Olfactory blockage experiment

### 2.3.1. Experimental set-up

To immobilize butterflies, a piece of paper was folded around both wings, while in upright/closed position, and secured in place with a bobby pin. Immobilization was conducted 1 h after eclosion, allowing wings to inflate and cuticle to harden before being handled, whereafter butterflies were haphazardly assigned to one out of three treatment groups.

### 2.3.2. Treatments

*Olfactory blockage group.* Unilateral blocking of olfactory input was accomplished by impeding the olfactory receptors of antennae, as we were concerned that surgical removal may lead to necrosis of the involved neurons. Pure beeswax was contained in a glass vial and kept just above the melting temperature of the wax, at about 65°C. The right-side antenna of each butterfly was transiently dipped into the vial, covering it with wax and thereby blocking olfactory reception without damaging associated olfactory receptor neurons.

*Control group.* After immobilization, animals with intact antennae were exposed to the same odour-rich environment as the olfactory blockage group.

*Newly eclosed group.* Brains of animals were dissected directly after immobilization, at a post-eclosion age of 2–3 h.

Animals of the olfactory blockage group and the control group were placed in individual plastic cups, with their right sides facing downwards, at 23°C for 7 days. Each cup was equipped with a cotton wad saturated with 25% sugar water for feeding. To provide an odour-rich environment, the treatment room was equipped with a large quantity of potted stinging nettles (*U. dioica*) (approx. 150 stems), six potted thistles in full bloom (*Cirsium arvense*) and four potted non-blooming sunflower plants (*Helianthus* sp.). Plants were kept at a distance of approximately 50 cm from the butterflies and continuously replaced to keep the odour profile consistent for the duration of the experiment. Animals were transferred to the odour-rich environment immediately after immobilization (control group) or treatment (olfactory blockage group).

## 2.4. Immunohistochemistry, scanning and reconstruction

After treatments, butterflies were decapitated and brains dissected and fixed. Methods for fixation, immunohistochemistry, confocal scanning and reconstruction were identical to a previous study [8], except that all samples were scanned from both anterior and posterior sides to maximize the clarity of morphological details in the investigated neuropils. Furthermore, as there is negligible variation in brain volume between individuals of the same age [8], absolute neuropil volumes were used in the analyses.

## 2.5. Statistics

All statistical analyses were performed using GraphPad Prism v. 8.0.2 for macOS. All neuropil subgroups except the right hemisphere ALs of newly eclosed butterflies adhered to Gaussian distributions. Computations including the subgroup which did not have a normal distribution of its values (Anderson–Darling test and Shapiro–Wilk test: $p > 0.05$) were done using non-parametric tests, while all other computations were done using parametric tests.

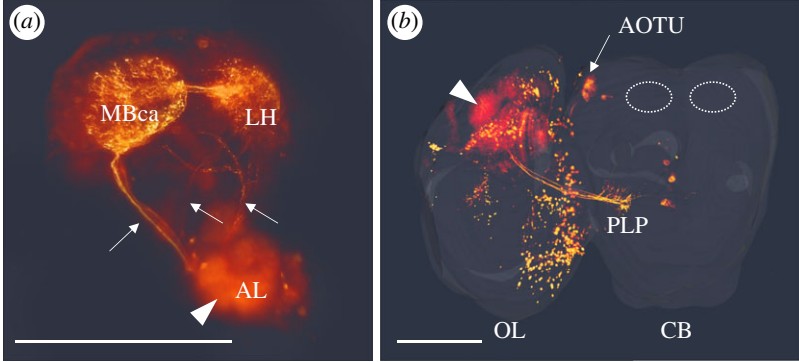

**Figure 1.** NB injection tracings showcase inter-neuropil connections and neural branching patterns in the brain of *P. c-album*. (*a*) Injections into the AL reveal three major tracts running from the AL to the MB calyx (MBca) and the LH. The arrowhead shows the injection site, and the arrows indicate the three tracts. Scale bar, 500 µm. (*b*) Injections into the lobular complex (arrowhead) of the optic lobe (OL) reveal a major tract terminating in a region in the posterior lateral protocerebrum (PLP) and also stain the anterior optic tubercle (AOTU). The calyces of the MB are depicted by the dashed ovals. The central brain (CB) and the left OL are shown as surface reconstructions of anti-synapsin-stained neuropils. Scale bar, 500 µm.

## 3. Results

Our main results were (i) innervation to the MB calyx was primarily olfactory, (ii) AL expansions were dependent on olfactory input, but independent of age, (iii) lack of olfactory input from one antenna prevented expansions of ipsilateral ALs and greatly reduced the expansion of contralateral ALs, and (iv) MB calyx expansions were age-dependent but independent of olfactory input.

### 3.1. Tracing experiment

To investigate the sensory innervation to the MB calyces, we performed a tracing experiment where NB was injected into either the optic sublobes (medulla or lobula/lobular plate), the MB calyx or the AL. Injection into either the medulla or the lobula complex did not show any axonal branching into the MB calyx (figure 1*a*) (but to visually oriented regions of the brain, e.g. the anterior optic tubercle). Neither did we observe any innervation in the optic lobes when injecting NB into the calyx (not shown). However, NB-injected ALs show one thick tract of AL output neurons targeting the MB calyx and projecting further to the LH (figure 1*b*). Furthermore, at least two thinner tracts connect the AL with the LH and continue to the MB calyces. Also, NB injection in the MB calyx demonstrated strong fluorescent staining in the AL and the LH (not shown). The innervation pattern of olfactory output neurons shown here with NB corroborates earlier stainings with Rhodamin dextran in this species [20].

### 3.2. Olfactory blockage experiment

#### 3.2.1. Antennal lobes

AL volume at eclosion was on average $9.1 \times 10^5 \, \mu m^3$ (s.e.m. $= 3.3 \times 10^4$; $n = 12$ animals) with no difference between hemispheres (Wilcoxon matched-pairs signed-rank test: $p = 0.85$; $n = 12$). While the bilateral symmetry observed at eclosion was preserved in the 7-day-old control group (paired *t*-test: $p = 0.28$; $t = 1.31$; d.f. $= 11$), AL volumes had significantly increased (Kruskal–Wallis test: $H = 33$; $p < 0.0001$) to an average of $14 \times 10^5 \, \mu m^3$ (s.e.m. $= 4.4 \times 10^4$; $n = 12$ animals), representing a 52% increase in total AL volume.

There was a significant difference in AL volume between treated and untreated hemispheres in the olfactory blockage group (paired *t*-test: $p < 0.0001$; $t = 9.2$; d.f. $= 16$). ALs of the untreated side were on average 37% larger than those of the treated side ($11 \times 10^5 \, \mu m^3$ versus $8.1 \times 10^5 \mu m^3$, s.e.m. $= 4.9 \times 10^4$ and $4.3 \times 10^4$, respectively, $n = 17$ animals). There was no significant difference in AL volume between the treated side of animals in the olfactory blockage group and either AL of newly eclosed animals (Kruskal–Wallis test: $H = 4.3$; $p = 0.12$). There was, however, a significant difference in AL volume between the untreated hemisphere in the olfactory blockage group and both ALs of the control group (ANOVA: $F_{2,38} = 7.7$; $p < 0.01$), representing a reduction in post-eclosion expansion by 56% compared to the control group.

Both ipsilateral and contralateral AL volumes were negatively affected by the lack of olfactory input from antennae which were treated with wax. As ALs ipsilateral to treated antennae did not at all increase in volume with age, the observed increase in AL volumes of the other subgroups appears to be dependent solely on olfactory input.

### 3.2.2. Mushroom body calyx

MB calyx volume at eclosion was on average $3.5 \times 10^5 \, \mu m^3$ (s.e.m. = $1.9 \times 10^4$; $n = 12$ animals), with no difference in volume between hemispheres (paired $t$-test: $p = 0.86$; $t = 0.19$; d.f. = 11). While the bilateral symmetry observed at eclosion was preserved in the 7-day-old control group (paired $t$-test: $p = 0.32$; $t = 1.0$; d.f. = 11), calyx volumes had significantly increased (ANOVA: $F_{3,44} = 12$; $p < 0.0001$) to an average of $5.2 \times 10^5 \, \mu m^3$ (s.e.m. = $1.9 \times 10^4$; $n = 12$ animals), representing a 48% increase in total calyx volume.

Calyx volumes of the olfactory blockage group were on average $5.2 \times 10^5 \, \mu m^3$ (s.e.m. = $2.0 \times 10^4$; $n = 17$ animals), and there was no difference in volume between hemispheres (paired $t$-test: $p = 0.45$; $t = 0.77$; d.f. = 16)). Calyces of this group were significantly larger compared to newly eclosed animals (ANOVA: $F_{3,54} = 10$; $p < 0.0001$), but there was no significant difference in volume between calyces of the olfactory blockage and control groups (ANOVA: $F_{3,54} = 0.13$; $p = 0.9$).

As there was no difference in calyx volume between treated and untreated hemispheres in the olfactory blockage group, nor any difference between olfactory blockage and control groups, the observed increase in calyx volume appears to be independent of olfactory input.

## 4. Discussion

Our results show that passively received olfactory input does not affect MB calyx volume but greatly affects AL volume, indicating differential functional pathways for post-eclosion volumetric expansion of the primary and secondary olfactory neuropils in *P. c-album*. Importantly, these results also indicate that the MB calyx expansions observed in other studies after various olfactory-related experiences were most likely caused by higher-order neural processes, such as spatial or associative learning, rather than by the sensory input *per se*.

The MB calyx in *P. c-album* is primarily innervated by AL output neurons (figure 1). In many other insects, it has been shown that the calyx also is a target for visual PNs from the optic lobes [21,50–57]. However, despite injection into several sites in the optic sublobes, we did not observe any MB calyx innervation. This does, of course, not exclude the possibility that second-order downstream neurons transfer visual information to the calyces. Nevertheless, our results indicate that the bulk of the information relayed to the MB calyces in this species is olfactory.

Our results show that while MB calyces of *P. c-album* increase in volume with age, they are volumetrically unaffected by olfactory input (figure 2). This finding is rather surprising as there is ample evidence showing that MB calyces increase in volume as a response to various olfactory-related experiences such as plant interactions [8,21], social interactions [5,58,59] and foraging efforts [4,5,60]. While previous research generally has approached these topics by investigating cumulative effects of complex experiences, we set out to isolate the olfactory component from behavioural and cognitive elements. By doing so, we were able to quantify the effects of passively received olfactory input on neuropil size. We can conclude that such input does not affect calyx volume in this species, and our results indicate that experience-dependent calyx expansions are likely to stem from processes related to behavioural expressions or cognitive functions, rather than increased levels of sensory input.

It was recently shown that females of *P. c-album* that experienced a complex host and non-host plant environment had larger calyces than females that were kept in an environment with a more neutral odour profile [8]. Although that study did not take the absolute number of insect–plant interactions into account and instead focused on the effect of general environment, it has previously been shown that MB calyx volumes of the butterfly *Pieris rapae* are linked both to total number of interactions with any plant type and with number of specific host plant interactions [21]. In the latter study, it was also shown that calyx volume at emergence, as well as after plant experience, was correlated with greater proficiency in finding host plants. As our results demonstrate a lack of connection between passive odour exposure and increased calyx volume, we suggest that the increased calyx volumes previously observed in ovipositing *P. c-album* [8] and *P. rapae* [21] are likely to have been caused by, for example, learning or other types of neural processing related to interactions with the plants.

The bilaterally uniform expansion of calyces in both control and olfactory blockage groups, together with the lack of neuropil expansion in ALs ipsilateral to blocked antennae, confirms that age-related calyx

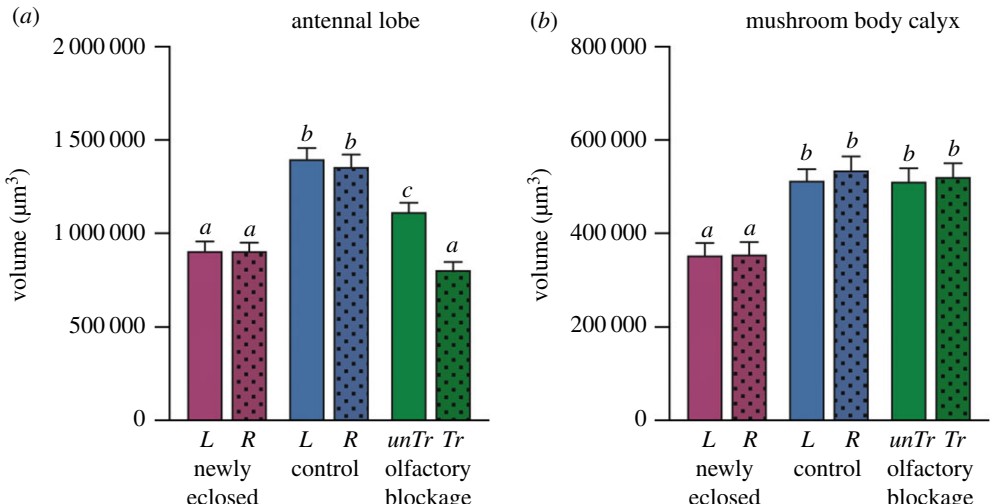

**Figure 2.** Bar graphs depicting absolute neuropil volumes, with error bars indicating s.e.m. Columns denoted with different letters are statistically different, dotted columns represent right hemisphere neuropils of newly eclosed (NE) ($n = 12$) and control (C) ($n = 12$) groups, and treated side of the olfactory blockage group (OB) ($n = 17$). L and R denote left and right hemispheres, and Tr and unTr denote treated and untreated hemispheres, respectively. (*a*) ALs of the NE and C groups were bilaterally symmetrical, while ALs of the OB group differed significantly in volume between hemispheres ($p < 0.0001$). ALs of the C group were significantly larger than those of the NE group ($p < 0.0001$), the treated hemisphere ALs of the OB group were of similar sizes as those of the NE group, while untreated hemisphere ALs reached volumes intermediate to NE and C. (*b*) Calyces of all groups were bilaterally symmetrical. There was no difference in calyx volume between C and OB groups, but both groups were significantly larger than NE ($p < 0.0001$).

expansions occur independently of olfactory input. Similar experience-independent but age-related expansions of MBs have been reported in several other insects [5,24,45,46,58,59]. Although it is notably difficult to disentangle effects of innate ageing from unspecified life experiences, the calyx expansion observed in the present study is in line with observations made for honeybees reared in darkness and social isolation [61], and *Drosophila* flies reared in darkness [6]. This experience-independent portion of MB expansions could be described as experience-expectant [61] if it provides the insect with increased neural processing power in anticipation of future life-history challenges.

The lack of experience-dependent calyx expansion in the present study does not exclude other types of structural changes to the neuropil. It is possible that the input had an effect on microglomerular density (MGD), synaptic proliferation, or other fine-scale structures, which were not investigated here. Fluctuations in MB calyx MGD have been observed in insects as an effect of age and various life experiences, they may occur with or without accompanying changes in neuropil volume, and appear to be of a transient nature [62]. Studies on honeybees and ants have shown that olfactory-related experiences coupled with a cognitive challenge can lead to increased calycal MGD without neuropils increasing in volume [63,64].

AL growth was arrested as a result of ipsilateral antennae being covered in wax (figure 2), indicating an absolute experience-dependence of AL post-eclosion expansion in this species while also confirming the treatment as successful in blocking olfactory input. The complete lack of expansion in ALs which were blocked from receiving input demonstrates that there is no innate ontogenetic growth in this neuropil during the first week after eclosion, and indicates that such experience-independent expansions later in life are unlikely. In fully functioning ALs, complex context-dependent sensory processing is likely to have an effect on neuropil volume when animals have the opportunity to act naturally upon behavioural cues. The ALs of bumblebees forced to forage in darkness, for example, are larger than those of bumblebees acting within a more natural foraging arena [24], indicating that ALs may increase in volume in insects which need to rely more heavily on olfactory cues as a compensation for visual impairment. In the light of the expansion induced by sensory input alone in our study, however, it is reasonable to assume that a significant portion of observed age-related AL expansions in other studies also would have been caused by sensory input, and that behaviourally induced expansions generally come at a lesser extent.

The finding that the development of the contralateral AL also was affected by olfactory blockage was unexpected (figure 2). In contrast with, for example, dipteran species, axonal branching of sensory

antennal neurons in lepidopteran species is purely unilateral [65]. However, we cannot exclude cross-talking between the ALs. In fact, *P. c-album*, just like most other insects, has a single bilateral neuron, expressing serotonin, which connects the two lobes with each other [20] and may be partly responsible for the bilateral interaction we observed.

Brain tissue is expensive to maintain [1] and the bigger the brain is, the more energy it consumes as every event of neural activity comes at a certain metabolic cost [66]. The inherent costs and benefits from having a large brain invariably lead to trade-offs between energetic investment into brain tissue and other bodily functions with high energy demands, such as reproduction. Thus, there are benefits to having a high degree of plasticity in traits related to unpredictable aspects of the environment, such as availability of mates, food and host plants. Investing in brain structures associated with recognizing host plants, learning novel hosts and memorizing the location of important resource hot-spots may increase fitness more than maximized egg production in habitats where resources are scarce, while the opposite would be true for resource-rich habitats. In this perspective, brain plasticity driven by the magnitude of cognitive challenges experienced by an individual seems logical. This particular trade-off has been exemplified through studies of the butterfly *Pieris rapae*, where cognitive ability in host search and MB volume are positively correlated with each other, but come at the cost of reduced fecundity [21,67].

## 5. Conclusion

We showed that direct olfactory input affects AL volumes, but has no effect on MB calyx volumes. We also showed that there is an absolute experience-dependence to the volumetric expansion of ALs as they do not increase in volume after adult eclosion when olfactory input is blocked. Our results indicate that experience-dependent volumetric expansions of MB calyces may be entirely dependent on internal processes, as opposed to sensory input, and occur only as a consequence of higher cognitive processes such as associative learning.

Data accessibility. Volumetric data is available from the Dryad Digital Repository at: https://doi.org/10.5061/dryad.dg35t5v [68].
Authors' contributions. M.E., S.N. and M.A.C. conceived the project, designed the experiments and contributed to writing and revising the manuscript. M.E. and M.A.C. prepared and performed the experiments. M.E. performed the statistical analyses.
Competing interests. We declare we have no competing interests.
Funding. The study was supported by a grant from Swedish Research Council (VR, 2015-04218) to S.N. and from Stiftelsen Olle Engkvist Byggmästare to M.A.C.

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
