## [Reviewer comments · Royal Society Open Science]

Review History

RSOS-190875.R0 (Original submission)

Review form: Reviewer 1 (Sylvia Anton)

Is the manuscript scientifically sound in its present form?

Yes

Are the interpretations and conclusions justified by the results?

Yes

Is the language acceptable?

Yes

Is it clear how to access all supporting data?

Yes

Do you have any ethical concerns with this paper?

No

Have you any concerns about statistical analyses in this paper?

No

Recommendation?

Accept with minor revision (please list in comments)

Comments to the Author(s)

The paper by Eriksson et al. presents highly interesting data on the influence of olfactory input on central neuropil volumes in a butterfly, by separating the influence of sensory input from behavioural and cognitive activity. The authors convincingly show that AL volumes are influenced by antennal sensory input, whereas mushroom body calyces, serving as secondary olfactory centre, are not. They show thus that earlier described experience-dependent changes in mushroom body calyces are most likely not due to sensory input alone, but caused by cognitive or behavioural activity.

The paper is very well written and illustrated and I only have a few minor comments as detailed below.

Line 54: ...olfactory cues...

Line 136: what do you mean with "simultaneously filled"? Do you mean that individual insects hatching at the same time were randomly attributed to one of three experimental groups?

Material and methods: it took me a while to understand what the difference between control and baseline group was. The easiest thing to clarify this would be to simply change to control group (instead of groups) in line 153 (or include age for each group to clarify?)

Line 183/84: Your point 3 is incomplete: I would mention that lack of olfactory input prevented expansion of the ipsilateral AL and reduced the expansion of the contralateral AL significantly.

Line 195/196: and projecting further to the lateral horn?

Line 198: "expression" is misleading, rather use something like "strong staining"?

Result part on antennal lobes page 11: You don't mention in the text, that also the contralateral AL of the blockage group is significantly smaller than the AL in control insects. As you discuss that point later, it would be good to mention this here already (even if it is mentioned in the figure legend).

Line 243: ...appears to be...

First paragraph on page 15 (Lines 306-308): you might want to mention that MGD density and MB volume can also decrease when social insects live in confined conditions (reviewed in Groh and Meinertzhagen, Front Biosci 2:268-288, 2010).

Line 573: ...are shown...

Line 582: ALs of the C group were significantly...

Line 583/84: ...the treated hemisphere ALs of the OB group were of similar size as those of the ...

Review form: Reviewer 2**Is the manuscript scientifically sound in its present form?**

Yes

Are the interpretations and conclusions justified by the results?

Yes

Is the language acceptable?

Yes

Is it clear how to access all supporting data?

Yes

Do you have any ethical concerns with this paper?

No

Have you any concerns about statistical analyses in this paper?

No

Recommendation?

Accept with minor revision (please list in comments)

Comments to the Author(s)

This study shows convincingly that the mechanisms underlying the growth of antennal lobes and mushroom bodies differ, and that olfactory input seems a main factor of AL development during early adulthood of the Comma butterfly. In my opinion, there are a few points that should be addressed:

1. Some aspects of the Methods section should be clarified or completed. For example, were the animals used for axonal tracing the same as those of the 'baseline group' in the second experiment? If not, how old were they? In the Results and figures, the 'baseline group' is actually referred to as 'newly eclosed' animals: please use only one name for this group throughout the manuscript. In that same experiment, it is said that the control group underwent no further treatment following immobilization, but it did as it was exposed to the same olfactory context as the treated one. Finally, what was the delay between wax treatment and exposure to the olfactory context?

2. The sample size differs, within the 'olfactory blockage' group, for measurements of AL and MB volumes (18 and 20, respectively), and as shown by the raw data, in some brains only the ALs were measured, and only the MBs in others (probably due to technical reasons). This might introduce a possible bias, as comparing the growth patterns of ALs and MBs should be based on measurements performed in the same brains. I suggest to exclude the brains whose analysis was incomplete from the dataset.

3. Olfactory blocking has a specific meaning in the field of behavioural studies (see e.g. Guerrieri et al 2005), so I recommend avoiding its use in the present study. I suggest 'blockade of olfactory input' or 'olfactory blockade' instead.

l.80: It is not clear to me why the authors mention the calyx as the input region and the lobes as the main processing region – the lobes are the main output region, and there is quite some processing occurring in the calyces, so unless I am missing strong arguments to maintain this dichotomy, I'd rather oppose the calyx and lobes as input and output regions, rather than involved or not in information processing.

l.277: what is a 'neutral' environment? Do you mean less complex? Comparatively poorer in terms of sensory stimuli? Please clarify.

l. 306-310: the 'cognitive challenge' in the cited studies was the establishment of an olfactory associative memory. In the present experiment, it is likely that butterflies have established such a memory as they were fed while smelling specific plant odorants during several days, even if they were immobilized (which was actually the case in the bee study). Thus, the hypothesis that MGD would not be affected in this experiment seems to lack strong support, while the opposite seems more likely. I suggest to modify that part of the manuscript accordingly.

l.316-317: why do the authors consider that the lack of AL expansion on the treated side indicate that expansion later in life is unlikely? I fail to see the logic, here - although it is clear that in many insect species the first days or week of adult life is a critical period for experience-dependent neuropilar growth (see e.g. Groh & Meinertzhagen, 2010; Cabirol et al. 2017)

Fig. 1: Please mention in the legend that the arrows indicate the three tracts.

Fig.2: Please indicate, in the legend, the sample sizes and statistical significance of letters (a,b,c)

Decision letter (RSOS-190875.R0)

19-Jun-2019

Dear Dr Carlsson

On behalf of the Editors, I am pleased to inform you that your Manuscript RSOS-190875 entitled "Insect brain plasticity: effects of olfactory input on neuropil size" has been accepted for publication in Royal Society Open Science subject to minor revision in accordance with the referee suggestions. Please find the referees' comments at the end of this email.

The reviewers and handling editors have recommended publication, but also suggest some minor revisions to your manuscript. Therefore, I invite you to respond to the comments and revise your manuscript.

- Ethics statement

- Data accessibility

If you wish to submit your supporting data or code to Dryad (<http://datadryad.org/>), or modify your current submission to dryad, please use the following link:
<http://datadryad.org/submit?journalID=RSOS&manu=RSOS-190875>

- **Competing interests**

- **Authors' contributions**

- **Acknowledgements**

- **Funding statement**

Because the schedule for publication is very tight, it is a condition of publication that you submit the revised version of your manuscript before 28-Jun-2019. Please note that the revision deadline will expire at 00.00am on this date. If you do not think you will be able to meet this date please let me know immediately.

When submitting your revised manuscript, you will be able to respond to the comments made by the referees and upload a file "Response to Referees" in "Section 6 - File Upload". You can use this to document any changes you make to the original manuscript. In order to expedite the

processing of the revised manuscript, please be as specific as possible in your response to the referees. We strongly recommend uploading two versions of your revised manuscript:

Kind regards,
Alice Power
Editorial Coordinator
Royal Society Open Science

on behalf of Dr Richard Benton (Associate Editor) and Kevin Padian (Subject Editor)
openscience@royalsociety.org

Reviewer comments to Author:

Reviewer: 1

Comments to the Author(s)

The paper by Eriksson et al. presents highly interesting data on the influence of olfactory input on central neuropil volumes in a butterfly, by separating the influence of sensory input from behavioural and cognitive activity. The authors convincingly show that AL volumes are influenced by antennal sensory input, whereas mushroom body calyces, serving as secondary olfactory centre, are not. They show thus that earlier described experience-dependent changes in mushroom body calyces are most likely not due to sensory input alone, but caused by cognitive or behavioural activity.

The paper is very well written and illustrated and I only have a few minor comments as detailed below.

Line 54: ...olfactory cues...

Line 136: what do you mean with "simultaneously filled"? Do you mean that individual insects hatching at the same time were randomly attributed to on of three experimental groups?

Material and methods: it took me a while to understand what the difference between control and baseline group was. The easiest thing to clarify this would be to simply change to control group (instead of groups) in line 153 (or include age for each group to clarify?)

Line 183/84: Your point 3 is incomplete: I would mention that lack of olfactory input prevented expansion of the ipsilateral AL and reduced the expansion of the contralateral AL significantly.

Line 195/196: and projecting further to the lateral horn?

Line 198: "expression" is misleading, rather use something like "strong staining"?

Result part on antennal lobes page 11: You don't mention in the text, that also the contralateral AL of the blockage group is significantly smaller than the AL in control insects. As you discuss that point later, it would be good to mention this here already (even if it is mentioned in the figure legend).

Line 243: ...appears to be...

First paragraph on page 15 (Lines 306-308): you might want to mention that MGD density and MB volume can also decrease when social insects live in confined conditions (reviewed in Groh and Meinertzhagen, *Front Biosci* 2:268-288, 2010).

Line 573: ...are shown...

Line 582: ALs of the C group were significantly...

Line 583/84: ...the treated hemisphere ALs of the OB group were of similar size as those of the ...

Reviewer: 2

Comments to the Author(s)

This study shows convincingly that the mechanisms underlying the growth of antennal lobes and mushroom bodies differ, and that olfactory input seems a main factor of AL development during

early adulthood of the Comma butterfly. In my opinion, there are a few points that should be addressed:

1. Some aspects of the Methods section should be clarified or completed. For example, were the animals used for axonal tracing the same as those of the 'baseline group' in the second experiment? If not, how old were they? In the Results and figures, the 'baseline group' is actually referred to as 'newly eclosed' animals: please use only one name for this group throughout the manuscript. In that same experiment, it is said that the control group underwent no further treatment following immobilization, but it did as it was exposed to the same olfactory context as the treated one. Finally, what was the delay between wax treatment and exposure to the olfactory context?
2. The sample size differs, within the 'olfactory blockage' group, for measurements of AL and MB volumes (18 and 20, respectively), and as shown by the raw data, in some brains only the ALs were measured, and only the MBs in others (probably due to technical reasons). This might introduce a possible bias, as comparing the growth patterns of ALs and MBs should be based on measurements performed in the same brains. I suggest to exclude the brains whose analysis was incomplete from the dataset.
3. Olfactory blocking has a specific meaning in the field of behavioural studies (see e.g. Guerrieri et al 2005), so I recommend avoiding its use in the present study. I suggest 'blockade of olfactory input' or 'olfactory blockade' instead.

l.80: It is not clear to me why the authors mention the calyx as the input region and the lobes as the main processing region – the lobes are the main output region, and there is quite some processing occurring in the calyces, so unless I am missing strong arguments to maintain this dichotomy, I'd rather oppose the calyx and lobes as input and output regions, rather than involved or not in information processing.

l.277: what is a 'neutral' environment? Do you mean less complex? Comparatively poorer in terms of sensory stimuli? Please clarify.

l. 306-310: the 'cognitive challenge' in the cited studies was the establishment of an olfactory associative memory. In the present experiment, it is likely that butterflies have established such a memory as they were fed while smelling specific plant odorants during several days, even if they were immobilized (which was actually the case in the bee study). Thus, the hypothesis that MGD would not be affected in this experiment seems to lack strong support, while the opposite seems more likely. I suggest to modify that part of the manuscript accordingly.

l.316-317: why do the authors consider that the lack of AL expansion on the treated side indicate that expansion later in life is unlikely? I fail to see the logic, here - although it is clear that in many insect species the first days or week of adult life is a critical period for experience-dependent neuropilar growth (see e.g. Groh & Meinertzhagen, 2010; Cabirol et al. 2017)

Fig. 1: Please mention in the legend that the arrows indicate the three tracts.

Fig.2: Please indicate, in the legend, the sample sizes and statistical significance of letters (a,b,c)

Author's Response to Decision Letter for (RSOS-190875.R0)

See Appendix A.

Decision letter (RSOS-190875.R1)

23-Jul-2019

Dear Dr Carlsson,

I am pleased to inform you that your manuscript entitled "Insect brain plasticity: effects of olfactory input on neuropil size" is now accepted for publication in Royal Society Open Science.

Kind regards,

on behalf of Dr Richard Benton (Associate Editor) and Kevin Padian (Subject Editor)
openscience@royalsociety.org

Appendix A

Dear Editor,

We are very grateful that you accepted our manuscript for publication in Royal Society Open Science. We have dealt with the comments and suggestions from the reviewers, which were very helpful, and made changes accordingly (see below). Furthermore, we have uploaded an updated manuscript (a clean version and one with tracked changes). The raw data has been uploaded and deposited in the Dryad Digital.

Best regards,
Mikael A Carlsson

Reviewer 1

Line 54: ...olfactory cues... *Fixed*

Line 136: what do you mean with “simultaneously filled”? Do you mean that individual insects hatching at the same time were randomly attributed to one of three experimental groups?

Yes, exactly. As the phrasing was a bit unclear, the two last sentences of that paragraph are now merged and we hope that the change will make it clearer as to what we want to express.

(Immobilization was conducted one hour after eclosion, allowing wings to inflate and cuticle to harden before being handled, and butterflies were then haphazardly assigned to one out of three treatment groups.)

Material and methods: it took me a while to understand what the difference between control and baseline group was. The easiest thing to clarify this would be to simply change to control group (instead of groups) in line 153 (or include age for each group to clarify?)

The s in groups on line 153 was there because we were referring to both the olfactory blockage group and the control group at the same time, but the sentence is now changed so that it is more clear which groups we really are talking about here. We also added the age at which animals of the baseline group were sacrificed on line 151/152. Please note that we also changed “baseline” to “newly eclosed” as reviewer 2 pointed out our inconsistent terminology.

Line 183/84: Your point 3 is incomplete: I would mention that lack of olfactory input prevented expansion of the ipsilateral AL and reduced the expansion of the contralateral AL significantly.

It was our intention that point 2 would make it clear that lack of olfactory input prevented expansion of ipsilateral AL, but we agree that your suggestion greatly improves clarity here.

Line 195/196: and projecting further to the lateral horn? *Yes, changed to “and projecting further to the...”*

Line 198: “expression” is misleading, rather use something like “strong staining”? *Changed “expression” to “fluorescent staining”*

Result part on antennal lobes page 11: You don't mention in the text, that also the contralateral AL of the blockage group is significantly smaller than the AL in control insects. As you discuss that point later, it would be good to mention this here already (even if it is mentioned in the figure legend).

It is mentioned in a sentence on line 221-224 that AL of the untreated hemisphere among olfactory blockage animals are smaller than the AL of control animals, but as it is quite hidden in the text we added a sentence to the final comment (line 226-228) on this result so that it now reflects the effects on both contra- and ipsilateral AL. We agree that it is better this way as it makes that part of the results more visible, thank you for bringing it to our attention.

Line 243: ...appears to be... *Fixed*

First paragraph on page 15 (Lines 306-308): you might want to mention that MGD density and MB volume can also decrease when social insects live in confined conditions (reviewed in Groh and Meinertzhagen, Front Biosci 2:268-288, 2010). *Thank you for bringing this fact and very nice paper to our attention. The whole subject of MGD is fascinating and there are many aspects which could have affected the MGD of our samples, but as we did not measure MGD we did not want to sidetrack the discussion too much. Although Polygonia c-album is not a particularly social species it is still possible, and perhaps even likely, that social interactions -or lack thereof- could affect their calycal MGD. However, we intentionally wanted this section of the discussion to be quite brief as to not divert too much focus into MGD and instead allow more room for discussing the physical aspects we did measure.*

Line 573: ...are shown... *Fixed*

Line 582: ALs of the C group were significantly... *Fixed*

Line 583/84: ...the treated hemisphere ALs of the OB group were of similar size as those of the ...*Fixed*

Reviewer: 2

Comments to the Author(s)

This study shows convincingly that the mechanisms underlying the growth of antennal lobes and mushroom bodies differ, and that olfactory input seems a main factor of AL development during early adulthood of the Comma butterfly. In my opinion, there are a few points that should be addressed:

1. Some aspects of the Methods section should be clarified or completed. For example, were the animals used for axonal tracing the same as those of the ‘baseline group’ in the second experiment? If not, how old were they? *No, they were not the same animals as the tracing and olfactory blocking experiments were two separate experiments, but females from the same population and age were used in both experiments, which is now clarified.*

In the Results and figures, the ‘baseline group’ is actually referred to as ‘newly eclosed’ animals: please use only one name for this group throughout the manuscript. *We used baseline throughout earlier versions of the manuscript and honestly just missed to change it in the methods section, we have now changed it to newly eclosed and greatly appreciate that you spotted this mistake so we could fix it.*

In that same experiment, it is said that the control group underwent no further treatment following immobilization, but it did as it was exposed to the same olfactory context as the treated one. *We agree that it is a bit confusing and have changed the sentence accordingly.*

Finally, what was the delay between wax treatment and exposure to the olfactory context? *The delay was minimal, just a few minutes, we added a sentence stating this at the end of that paragraph.*

2. The sample size differs, within the ‘olfactory blockage’ group, for measurements of AL and MB volumes (18 and 20, respectively), and as shown by the raw data, in some brains only the ALs were measured, and only the MBs in others (probably due to technical reasons). This might introduce a possible bias, as comparing the growth patterns of ALs and MBs should be based on measurements performed in the same brains. I suggest to exclude the brains whose analysis was incomplete from the dataset. *You are right, we have removed all individuals of which we were not able to reconstruct both ALs and calyces from the whole dataset, including two individuals from the control group where one only had measurements from the AL and the other only from calyces.*

3. Olfactory blocking has a specific meaning in the field of behavioural studies (see e.g. Guerrieri et al 2005), so I recommend avoiding its use in the present study. I suggest ‘blockade of olfactory input’ or ‘olfactory blockade’ instead. *Thank you for bringing this to our attention and we have changed blocking to blockade in all of the appropriate places, and hope that this is a satisfying solution.*

1.80: It is not clear to me why the authors mention the calyx as the input region and the lobes as the main processing region – the lobes are the main output region, and there is quite some processing occurring in the calyces, so unless I am missing strong arguments to maintain this dichotomy, I’d rather oppose the calyx and lobes as input and output regions, rather than involved or not in information processing. *You are right, we now changed it to output region instead of main processing.*

1.277: what is a ‘neutral’ environment? Do you mean less complex? Comparatively poorer in terms of sensory stimuli? Please clarify. *We do mean less complex/comparatively poorer in terms of olfactory stimuli, as those details are explained in the cited paper, but also realize that we should be more precise here. The sentence is now changed so it instead reads “...had larger calyces than females that were kept in an environment with a more neutral odour profile”. Thank you for bringing it to our attention.*

1.306-310: the ‘cognitive challenge’ in the cited studies was the establishment of an olfactory associative memory. In the present experiment, it is likely that butterflies have established such a memory as they were fed while smelling specific plant odorants during several days, even if they were immobilized (which was actually the case in the bee study). Thus, the hypothesis that MGD would not be affected in this experiment seems to lack strong support, while the opposite seems more likely. I suggest to modify that part of the manuscript accordingly.

We agree that we lack support that MGD would not be affected and we modified the text accordingly.

1.316-317: why do the authors consider that the lack of AL expansion on the treated side indicate that expansion later in life is unlikely? I fail to see the logic, here - although it is clear that in many insect species the first days or week of adult life is a critical period for experience-dependent neuropilar growth (see e.g. Groh & Meinertzhagen, 2010; Cabirol et al. 2017) *That is because we are talking about innate ontogenetic growth as opposed to growth triggered by sensory input. Our reasoning is that we find it unlikely that such growth would suddenly occur in females older than 7 days when it has been absent during the full first week after eclosion. As we stopped our measurements after 7 days we can of course not say with certainty that no innate ontogenetic growth would occur in older adults, but we fail to see a likely scenario where this would occur. If we were to reverse the blockade of olfactory input so that the antenna would start receiving olfactory input, however, we would expect growth to occur at any age. We would also expect ALs of untreated and fully functioning antennae to continue increasing in volume throughout the natural life of the butterfly, albeit probably at a slower pace in older butterflies.*

However, we appreciate your comment and have changed the sentence as to state more clearly that we are referring to innate experience-independent ontogenetic growth.

Fig. 1: Please mention in the legend that the arrows indicate the three tracts. *Fixed*

Fig.2: Please indicate, in the legend, the sample sizes and statistical significance of letters (a,b,c) *Fixed*